# The International Classification of Functioning, Disability and Health: Accuracy in Aquatic Activities Reports among Children with Developmental Delay

**DOI:** 10.3390/children10050908

**Published:** 2023-05-22

**Authors:** Merav Hadar-Frumer, Huib Ten Napel, Maria José Yuste-Sánchez, Isabel Rodríguez-Costa

**Affiliations:** 1Israel Sport Centre for the Disabled (ISCD) Ilan Spivak, Ramat Gan 52535, Israel; 2Faculty of Medicine and Health Sciences, University of Alcalá, 28807 Alcalá de Henares, Spain; 3Department of Primary and Community Care, Radboud University Medical Center, PB 9101, 6500 HB Nijmegen, The Netherlands; 4RIVM/Dutch WHO-FIC Collaborating Centre, PB 1, 3720 BA Bilthoven, The Netherlands

**Keywords:** the International Classification of Functioning, Disability and Health (ICF), the International Classification of Functioning, Disability and Health for Children and Youth Version (ICF-CY), aquatic activities, children aged 6–12, developmental delay

## Abstract

Functioning, as described in the International Classification of Functioning, Disability and Health (ICF), increasingly raises interest in the world of child rehabilitation, especially because its application empowers patients and parents by not putting the emphasis on disability in terms of the medical diagnosis but on the person’s lived experience and the level of functioning that might be achieved. However, this requires the correct understanding and application of the ICF framework to overcome differences in the often locally used models or the understanding of disability, including mental aspects. To evaluate the level of accurate use and understanding of the ICF, a survey was performed on studies of aquatic activities in children aged 6–12 with developmental delay published between the years 2010 and 2020. In the evaluation, 92 articles were found that matched the initial keywords (aquatic activities and children with developmental delay). Surprisingly, 81 articles were excluded for not referring to the ICF model at all. The evaluation was performed by methodological critical reading according to the ICF reporting criteria. The conclusion of this review is that despite the rising awareness in the field of AA, the ICF is used inaccurately and often not according to the model’s biopsychosocial principles. For the ICF to become a guiding tool in evaluations and goal-setting for aquatic activity, the level of knowledge and understanding of the framework and language needs to be increased via curricula and studies on the effect of interventions on children with developmental delay. Even more so, the level of understanding on how to apply functioning among instructors and researchers working in the aquatic environment needs to be increased.

## 1. Introduction

In the last few decades, the use of the aquatic environment (AE) has become a well-known and unique tool when it comes to restoring and promoting quality of life [1,2,3,4,5,6,7,8,9]. Many aquatic activities (AAs), focused on promoting human capabilities and rehabilitation, have been developed and studied under general names, such as “Hydrotherapy” and “Aquatic therapy” (and “swimming therapy”, “aquatic exercise”, “pool therapy” and more). These are aquatic-based activities conducted by an array of professionals, such as physical therapists, occupational therapists, speech therapists, physical education teachers, special education teachers, swimming instructors and more. These professionals utilize the aquatic properties (i.e., hydrostatic pressure, buoyancy, turbulence, viscosity, temperature, etcetera) along with special aquatic techniques and approaches developed specifically for the overall rehabilitation and/or sports and leisure activities of those undertaking aquatic activities [1,2]. As the growing awareness of this unique environment increases, so does the evidence on the success of AAs as a tool with a positive effect on various and varied areas of rehabilitation, with special emphasis on child habilitation and rehabilitation [1,2,3,5,6,7,8,9].

Working with children in the aquatic environment has been shown to be effective in promoting daily living activities and swimming skills [3,4,5,6,7,8,9,10]. In some cases, the aquatic environment is the only environment where individuals with severe limitations are able to move and practice active movement that cannot be practiced on land [6]. Furthermore, aquatic activities have been found to be effective in improving motor abilities, physical activity, social interaction, quality of life (QOL) and participation in children with developmental disabilities or delays [3,4,5,6,7,8,9,10]. In addition, game activities and swimming are very enjoyable and fun activities, allowing children to persist with therapeutic exercises with pleasure and motivation to persevere [2,3,6,7,8,10].

From the literary reviews conducted over the years on various studies in the field of AAs with children with developmental delays, the heterogeneity in the professional language, in the defined goals, in the measurements and in the conclusions rises again and again [10]. This fact makes it very difficult for researchers and professionals to reach similar conclusions and a uniform language that will make it possible to jointly promote the professional ability in the field of water activities with children [10].

The main motivation for the development of the ICF by the World Health Organization (WHO) was to create a common, nonjudgmental framework and language that would represent people from all over the world and enable the documentation and monitoring of the health conditions of children and adults [11,12,13].

The model provides clear definitions with neutral terms (without unnecessary negative connotation) [13]. It provides ways to describe a person’s problems through the assignment of codes and universal qualifiers. Assessment instruments linked to the ICF are being developed by the WHO with a view to applicability in different cultures [11,12,13].

For these reasons and since, as of today, there is no common denominator in the AA world that would unite the various goals, tools and interventions in order to create a similar language and unity among the many professionals, the ICF, developed by the WHO, especially for these purposes, is the right model for creating a common language and promoting activity and research in the AE [4,10].

### 1.1. The ICF Framework

The ICF framework was first introduced to the world by the WHO in 2001. It is a framework in which there are ongoing interactions between the health conditions, contextual factors (environmental and personal), functioning, disability and well-being of a person’s life [11,12,13,14,15,16]. The ICF is a “biopsychosocial” approach regarding the person’s functioning or disability within all aspects of life. The ICF model essentially deals with the quality of human life and well-being. In fact, the model describes and defines all the terms related to human health and some of the important aspects for its well-being.

The ICF model uses familiar terms with some new contexts and explanations. For example, the use of the terms ”impairment”, “disability” and “handicap” in the International Classification of Impairments, Disabilities and Handicaps was revised in the ICF framework; the term “handicap” was abandoned, and “disability” was used as an umbrella term for all three perspectives of the body, individual and societal [11]. For this reason, great emphasis is placed on ensuring that the professionals who use the model are well acquainted with the terms and use them appropriately. To this end, the model provides users with a glossary that is as broad and clear as possible, so it can provide a way of communication and a common language between all professionals and can facilitate collaboration among all those concerned with health [10,13].

#### 1.1.1. The ICF’s Main Terminology

Well-being (or quality of life)—a general term encompassing the total universe of human life domains, including physical, mental and social aspects. Well-being is a subjective feeling, i.e., what people feel about their health condition and its consequences on their life;Health condition—an umbrella term for an illness, disorder, injury or trauma as well as other circumstances, such as pregnancy, aging, stress, a congenital anomaly or genetic predisposition. HCs are coded using the International Classification of Diseases, Tenth Revision (ICD-10) [17];“Functioning” and “Disability”—two umbrella terms which are associated with health conditions. These two classifications are complementary and should be used together. Both terms encompass all the aspects of the interaction between the individual’s body functions, activities and participation and its contextual factors. “Functioning” represents all the positive aspects, while “Disability” represents all the negative aspects, such as impairments, activity limitations and participation restrictions [11,12,13,18,19];Qualifiers—the ICF framework offers means to assess a person’s functioning using universal numeric codes with values from 0 = no problem to 4 = complete problem. Their role is to specify the extent or the magnitude of functioning or disability within the BF and BS components of functioning. The other two qualifiers are related to theindividual’s environment while he/she is performing an activity (related to A&P). These codes define the individual’s “Capacity” (the highest probable level of functioning in a uniform or standard environment) and the “Performance” (what individuals do in their current environment). The EF qualifiers refer to the effect of the environment on functioning, i.e., whether they help (facilitators) or interfere (barriers) [11,12,13,18].

#### 1.1.2. The ICF Model—Interactions between the Components

The ICF diagram (Figure 1) represents the ongoing interactions between all components of the model [13,14,15]. By its definition, the ICF model is “a classification of people’s health characteristics within the context of their individual life situations and environmental impacts. It is the interaction of the health characteristics and the contextual factors that produces disability” [11] (p. 250). According to the ICF, a person’s functioning and disability in each of the various components (i.e., body function, body structure, activity and participation) always depend on a very complex interaction between his/her health condition and the person’s contextual factors—the environment factors and personal factors. A certain activity in a supportive environment will be considered functioning, while in a different environment, it can become a limitation [11,12,13,15,20]. In the same way, a certain activity performed by two people with different PFs can be perceived by one as functioning while by the other as a limitation.

As is shown by the bidirectional arrows in the diagram, the interaction works in two directions. There is no hierarchy; interventions in one entity have the potential to modify one or more of the other entities. For example, a health condition can affect functioning, while a change in functioning for the better or the presence of a disability can change the health condition itself [10,13]. The ICF model provides a multiperspective approach to the classification of functioning and disability as an interactive and evolutionary process. It “provides a standard language and framework to facilitate communication across services, organizations and agencies” [21] (p. 69), and by using the framework, one can identify the functioning abilities and decide on the intervention needed to track the status over time and assess the intervention’s outcome [21].

#### 1.1.3. The Use of the ICF in the Professional Literature

Since the launch of the ICF in 2001, every year, interest in the model has increased, and the number of publications related to the ICF has grown greatly. Researchers involved in the development and promotion of the ICF felt that in many of the articles, the understanding of the ICF is inadequate and may result in inaccurate use of the model [22]. In 2014, the WHO’s Family of International Classifications (WHO-FIC) Functioning and Disability Reference Group published a list of 12 criteria that they found to be “necessary in considering the quality and merit of the publication and its contribution to the overall ICF literature” [23].

In 2018, Daugaard et al. [22] validated these ICF reporting criteria and published them again to serve as guidelines for the purpose of “promoting transparent, clear and accurate reporting on the use of ICF” and in order to “assist researchers, editors and readers to identify quality publications on topics related to the International Classification of Functioning, Disability and Health (ICF)” [22]. The current list contains 11 questions to which the researcher should answer in order to determine whether publications are transparent, clear and accurate regarding their use of the ICF [24].

### 1.2. The Aim of this Scoping Review

The aim of the current review is to evaluate the use of the ICF in studies examining the effect of AAs on children (ages 6–12) with developmental delay by examining two subjects:The level of use of the ICF framework in general, i.e., the extent to which it is actually used in studies between 2010 and 2020;The level of mastery of the authors of the articles in their use of the ICF terms and the level of understanding of the framework itself, i.e., whether the ICF framework is fully understood and communicated accurately.

The study uses Daugaard and associates’ [22] ICF reporting criteria guidelines, which were developed for these purposes.

## 2. Materials and Methods

### 2.1. Study Design

A scoping review.

### 2.2. Search Strategy

Article selection: studies published in the time period from 1 January 2010 until 31 January 2020, which investigated the effect of AAs on children with developmental disability at elementary school age (6–12 years), were collected;The search was limited to studies that were published in English, full articles and open to the public on the Internet or in the medical libraries of Alcala University, Ben Gurion University in the Negev, Tel Aviv University and Sheba Medical Center;Electronic databases: Relevant articles were identified by searching among the international healthcare databases PubMed, PubMed Central^®^ (PMC), Google Scholar, Physiotherapy Evidence Database, Cochrane Library, ResearchGate, Scientific Research and Scielo. The search also reviewed the bibliographic references of the collected papers for the purpose of locating additional studies not found in the basic databases;Keyword combinations used for the search were the term “A child/children” with all terms related to aquatic activities and accompanying the following concepts: “hydro”, “aquatic”, “pool”, “swimming” and “water” (Table 1);The selection process of the articles contained two stages: at the first stage, the researchers looked for the criteria word or combination (Table 1) in the title and the abstract. If it was found to be matching, the next stage of evaluating the full text was performed. After these stages, both reviewers (MHF and IRC—both physical therapists and aquatic therapists for many years) debated about disparities, which were cleared up after re-examination of the full text and discussions about them. No rerun work was carried out prior to the final analysis.

### 2.3. Inclusion and Exclusion Criteria

The main evaluation areas that were defined were the following areas:Studies characteristics: All articles should have been published within the defined time period. The articles should contain all the details of the research, including the full results. The articles types were: descriptive research, a systematic scoping review, a literature review, an intervention review, an experts’ opinion article, a consensus process to report, narrative review and quasi-experiment or an integrative review;Population: The main population of the studies should be children with developmental delays aged 6–12. Studies that included different age groups were also included as long as this age group was included in the study. For example, in the review by Gorter and Currie, 2011 [25] of 6 previous articles, out of the 40 children participating in the various studies, 30 matched the age group defined in the current review;Aquatic methods used in the interventions: The focus is on studies that examined the effect of AAs without aids, such as floats, special seats and more. The types of intervention included different aquatic activities, such as swimming, aquatic therapy or any other physical activity in the aquatic environment, individually or in groups. The techniques used by the instructors, the means of instruction and the nature of accessibility to the children, as well as the environment in which the intervention took place, were different and diverse. In the world, AAs are widely used by many professionals, each referring to the activity under a different category, hence the need to expand the number of categories in order to include in the review as many types of AAs as possible;Relation to the ICF: the article should refer to the ICF, whether in the form of an explanation, a link to the research topics or in the results, as described in Daugaard et al.’s [22] guidelines on ICF reporting criteria;Quality assessment: The evaluation of the articles was carried out by a review reading, including a data collection and analysis process in which all the criteria defined in the guidelines of Daugaard et al. [22] were scanned and summarized for each and every article. A discussion was held between the ICF expert (HTN) and the lead author (MHF), and the final conclusions were reached after a procedure of agreement between the two researchers.

## 3. Results

### 3.1. Electronic Search Results

The first electronic screening ended up with 155 papers that met the initial criteria. The second screening process included reviewing all the articles found and a selection of eligible articles based on the inclusion/exclusion criteria, which included the study characteristic, population (age and diagnosis), language and aquatic methods. After reviewing all the papers, 64 were excluded, and 91 were included for the next step of analysis. In the third screening, all articles that did not relate to the ICF were excluded; 80 articles were excluded, and 11 papers were left. At the fourth screening, two more articles were excluded as the ICF model was not defined by the researchers as an important subject in their research. Nine papers were included in the final analysis (Figure 2).

Out of the 9 selected articles, 5 were various types of review articles which together reviewed 51 articles. Among the different reviews, there was an overlap of eight articles that were reviewed in two or three articles.

### 3.2. Articles Included

The following Table 2 shows the main data of the articles selected for review.

### 3.3. Quality Analysis—The ICF Reporting Criteria

The following Table 3 shows the summary of the findings organized according to the criteria of Daugaard et al. [22].

### 3.4. An Expansion on the Results of the ICF Critical Reading Presented in Table 3

From the results of the quality analysis according to the ICF reporting criteria, it is visible that all studies indeed use the ICF model as a tool and as a language. At the same time, it was found that the language is not consistent. Sometimes, the terms used by the authors are incorrect or inaccurate, and, in most studies, important basic areas of the ICF principles have been omitted.

A focused explanation of the results shown in Table 3 is presented below:A.The ICF’s language (criteria 2, 4 and 5)—term definitions and interactions: All nine articles were published ten years or more after the advent of the ICF model in the world. In most of the articles, the introduction of the model, its terms, domains and categories is very short or missing (except for the articles of Gorter and Currie [25], Cross et al. [10] and Declerck [20]). The vast majority of authors seem to presuppose that the model’s structure and principles are clearly understood by the readers and tend to skip explanations of the terms and their role as tools for evaluating functioning and disability.Another point is that there is an inconsistency in the language of the model. Some authors replace the definition of “components” with the word “levels” (Blohm [26]) or “categories” (Gorter and Currie [25]), and the term “domain” is also replaced by the term “category” (Cross et al. [10]; Declerck [20]) and “area” (Khalaji et al. [28]). Another misapplication is of the word “Disability” as a term that describes a health status, e.g., “Cerebral Palsy is the most common motor disability” [20] (p. 1) and “children with disabilities” [10] (p. 6);B.Addressing the various components of the ICF (criteria 1 and 9): In this area, there is a noticeable omission to the unique important factor added in the ICF model—the contextual components and especially the PF. Only two articles addressed the contextual components specifically while explaining the model (Gorter and Currie [25] and Declerck [20]). These two articles are also the only articles that refer to the PF as an important component in the children’s lives for future studies [25] and as an important factor to look for in the studies’ outcomes [20]. In all other articles, the PF is mentioned incidentally or not mentioned at all;C.The ICF qualifiers (criteria 7 and 8) The vast majority of the authors ignore the ICF qualifiers. In four out of the nine studies, various ICF qualifiers are mentioned to a small extent (Gorter and Currie [25]; Güeita-Rodríguez et al. [4,29,30]). No article used the qualifiers in the research;D.Awareness of the literature predating the study and relevant reference to the ICF literature (criterion 3): As expected, all nine articles addressed previous research and publications. Eight out of nine also linked them to various components of the ICF. Only one (Khalaji et al. [28]) did not refer to the ICF literature at all. One surprising fact is that Khalaji et al. [28] did not refer to ICF literature at all even though the study was looking for researches on hydrotherapy and its application for the improvement of the ICF in spastic diplegia cerebral palsy patients;E.The ICF linking to another tool and ICF-based instruments, including previous articles (criteria 3, 6 and 10): Only one group of authors - Güeita-Rodríguez et al. [4,29,30]—provided a full description of the recommended methodology (e.g., linking research outcomes to the ICF components). Some authors gave an example of links between the results of the studies reviewed and the ICF components but did not explain the methodology used [10,25,26,27]. Most authors did not mention the linkage process at all [25,29].Regarding the relationship between the ICF and the tools that were described in the articles, there is little more information on this subject. In five of the articles, the researchers explain the linkage made between the ICF components and the ICF-based tools that were developed in order to allow researchers to provide measures of functioning (Cross et al. [10]) or quality of life (Declerck [20]) and therapeutic intervention tools, questionnaires and APT-CSs (Güeita-Rodríguez et al. [4,29,30]).F.Knowledge translation between different settings (criterion 11): The analysis of the studies found that six of the articles refer to the ICF model as a language that creates a connection between the various assessment tools and enables the construction of a common base of knowledge between the various professions [4,10,20,27,29,30]. It can also serve as a framework that addresses the aquatic environment and group activities as two EFs that constitute unique factors that affect the child’s quality of life, functioning and motivation (Sutthibuta [27]). Güeita-Rodríguez et al. [4,29,30] developed preliminary APT-CSs for children and youth with neurological disorders. All data collection and consent work were based on a combination between the ICF-CY model, the WHO methodology for the ICF’s core sets (ICF-CSs) [31,32], the principles of the Rehab-Cycle model [33] and the Delphi technique [34].

## 4. Discussion

The purpose of the critical reading according to the ICF reporting criteria guidelines [22] is to assess whether the selected articles do indeed present the definitions and goals in an accurate way and in accordance with the framework of the ICF model.

As stated in this paper, the ICF reporting criteria guidelines’ goal is primarily to help authors improve articles related to the ICF framework by using a set of guiding questions that allow them to test whether the level of knowledge presented in the article is accurate, clear and provides appropriate reporting on the use of the ICF [22].

This discussion will focus on the main issues examined according to the ICF guidelines [22] and which were found in the data analysis of the selected studies:A.The ICF’s language (criteria 2, 4 and 5)—Term definitions and interactions: The complexity of the model requires a different perception than the traditional medical model, which views disability as a problem of the person caused by his/her health condition and to be managed by medical care. It also differs from the social model, which views disability as a complex collection of conditions, mostly created by the social environment where the management of the problem requires social action. The ICF is more than an integration of these two opposing models. It provides a “biopsychosocial” approach regarding the person’s functioning or disability [11,12,13]. The model is a globally agreed-on conceptual framework and common language for health purposes [35]. The change in terms and the addition of contextual factors requires more in-depth consideration and the provision of more details to create a sufficient level of knowledge among readers and professionals at a level that will allow a uniform, international, interprofessional language. According to the principles of the ICF framework, the various health components do not stand on their own. Each of the different components has multiple interactions with the other components and can affect and be affected by them. For this reason, in any reference to the model, it is necessary to give place to the interactions and effects, explain them and look for the interactions between the various components.As for the inconsistency in the language of the model, the ICF framework is a new professional language that uses a mix of known and new concepts and terms in a special way that is explained in the model. Each ICF component (except for personal factors) consists of various domains (chapters), and within each domain, there are categories, which are the wording units of classification that enable the assessors to select the appropriate health and health-related states of an individual. The ICF terminology is important in the creation of an international uniform language [11,12,13,15,20].As for the misapplication of the word “Disability” as a term that describes a health status, previously, when the classification model was based on a medical diagnosis, the word “disability” was defined as a problem of the person caused by his/her health condition and required medical care aimed at cure or helping the patient to adjust. Within the ICF model, disability, similarly (but opposite) to functioning, is an umbrella term for impairments, activity limitations and participation restrictions and results from the interaction between the person’s health condition and his/her contextual factors. By defining disability as an umbrella term, the ICF model acknowledges that every human-being can experience a decrement in health and thereby experience some disability. It shifts the focus from the health condition to the context, such as personality, past experience and the situation (PFs and EFs), which can be factors that promote functioning and, to the same extent, may cause disability in a particular situation [11,12,13,15].Only three articles [10,20,26] explicitly addressed the interaction between the ICF components indicates a deficiency in the overall broader reference to the model. Among the articles referring to the interactions, Sutthibuta [27] refers to interactions only within the social aspect. This reference is also incorrect, as the ICF model does not separate the various factors that affect the functioning and disability of the individual. By definition, all factors, including the social factors, have an influence on the child’s functioning and should be taken into consideration [11,12,13].One important interaction that professionals should recognize and note is the aquatic environment. The AE provides new opportunities for various limiting physical, social and emotional conditions, so professionals should be aware of its benefits when looking at the overall changes in the child’s functioning;B.Addressing the various components of the ICF (criteria 1 and 9): The fact that the two contextual factors are the components that have been left out in most of the studies is very thought-provoking, as these two components constitute one of the essential changes that the ICF model represent, and in fact, these are the newly added components in the classification [11,18]. According to the ICF framework, “Contextual Factors represent the complete background of an individual’s life and living” [13] (p. 15). EFs are extrinsic to the individual; PFs, on the other hand, are intrinsic (and not classified in the current version of the ICF). These factors have always been part of every person’s life and somehow were ignored until the ICF was published [16]. However, without the personal and environmental background of each person, it is not possible to really understand his/her functioning and the connections between the various components of the ICF in the context of that person. It is also important to notice that the concept of “quality of life” is often associated with the ICF as one of the PF domains, as estimating quality of life is primarily a subjective issue [19,36,37].Evaluation and classification according to the ICF framework without information about the PFs and EFs of the person are deficient and do not faithfully reflect the person’s functioning [19,38]. This fact is of particular importance when investigating children’s abilities and behavior in the AE. The aquatic atmosphere is a unique environment that changes motor control ability and reduces the control of gravity over the body [10,20,23]; therefore, children require a special emotional and physical adjustment (PF) for them to act in it and cooperate;C.The ICF qualifiers (criteria 7 and 8): The qualifiers of the ICF model are very important. Their role is to enable an assessment of the range or size of functioning or the disability and the changes within the various categories of functioning and the environment. Without the qualifiers, the ICF classes have no meaning. The qualifiers make it possible to compare between the current situation and improvement or deterioration in the future. This is why when we do any assessment, the codes should be accompanied by qualifiers. [11,12,13,14,21,38]. Future articles and studies should address these issues in the same way as addressing all components and domains of the ICF, especially when linking intervention outcomes to ICF components;D.The ICF linking to another tool and ICF-based instruments, including previous articles (criteria 3, 6 and 10): The linking process provides researchers with the ability to analyze research results in terms of description, comparison, quantitative data collection and more [39,40]. The process by which it is recommended to make the link between research results and the components of the ICF is described by Cieza et al. [40,41] and discussed in Fayed et al. [42]. To achieve the desired global change and to make the ICF the key factor in clinical use for rehabilitation purposes and studies, the ICF should be part of all studies, and researchers should use linking rules for the purposes of understanding the measures and to relate them to the ICF. To do so, researchers have to agree on linkages and develop versions of currently used instruments based-on the ICF (such as the work of Güeita-Rodríguez et al. [29]), which examine all the ICF categories and domains of the individual [32,43,44,45,46,47];E.SummaryFrom the analysis of the articles, one can see that most of the authors focused mainly on the importance of the ICF framework as a common language. Unfortunately, there is a very small focus on the other area of the ICF model as a tool for evaluating functioning and disability and for monitoring progress in aquatic rehabilitation procedures (as defined in the model objectives [11,12,13]);F.Recommendations for future research arising from this review:1.It is important to use the model as a whole, using the appropriate terminology and without omitting various components, which is disruptive to the holistic approach of the framework;2.Researchers should be aware of the possibilities that exist in the ICF model as an evaluation tool for research interventions. Previous studies [48,49] have demonstrated the use of this model for the purposes of evaluating children as an important factor for a holistic view of the child, as a tool that enables systematic data collection and broad information and as an interprofessional language;3.It is very important in future studies to use uniform and selective tools whenever linking the results of the various studies to the ICF fields;4.From the current analysis and the conclusions of Nguyen et al. [16] arises an inference regarding the importance of developing orderly models that will make it easier for professionals to use the ICF as a clinical tool for setting treatment goals and indices, i.e., a tool that will use the ICF framework’s qualifiers as an important part of the evaluation system. In Appendix A, our recommendations for promoting the accessibility and use of the ICF model in AAs and research are detailed.

## 5. Conclusions

This scoping review was focused on two main subjects: the level of the use of the ICF framework in studies between 2010 and 2020 and the level of mastery of the authors within the ICF framework.

The research findings main conclusions are:This systematic review, based on the ICF reporting criteria, showed that despite the rising awareness in the field of AAs, the ICF is applied inaccurately and most often not according to the model’s biopsychosocial principles. This inaccurate and incomplete application of the ICF hampers the comparability of research and further development of aquatic activities on an international level;Knowledge and understanding of the model are still lacking for some researchers, as expressed in the articles by confusion between the concepts or ignoring some of the model’s components. In order for the ICF to become a guiding tool in research for the purpose of evaluations and setting goals for AAs, it seems that there is a need for broader training programs for professionals in the field of AAs, as well as in-depth familiarity with the ICF model and its goals and applications in order to allow professionals to promote and perfect their abilities using the model.

## Figures and Tables

**Figure 1 children-10-00908-f001:**
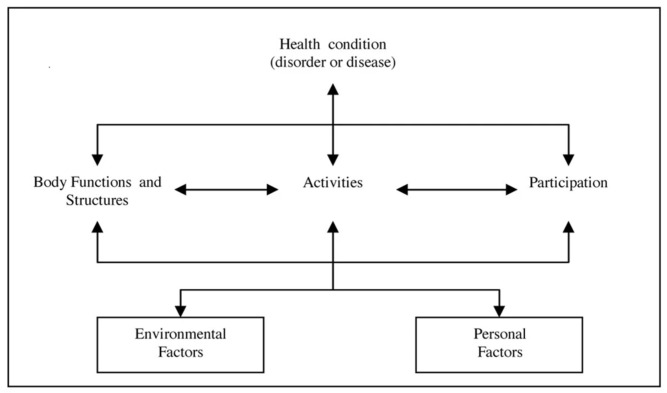
The ICF framework: Interaction between ICF components. (Adapted from Ref. [11]. Copyright 2001 WHO).

**Figure 2 children-10-00908-f002:**
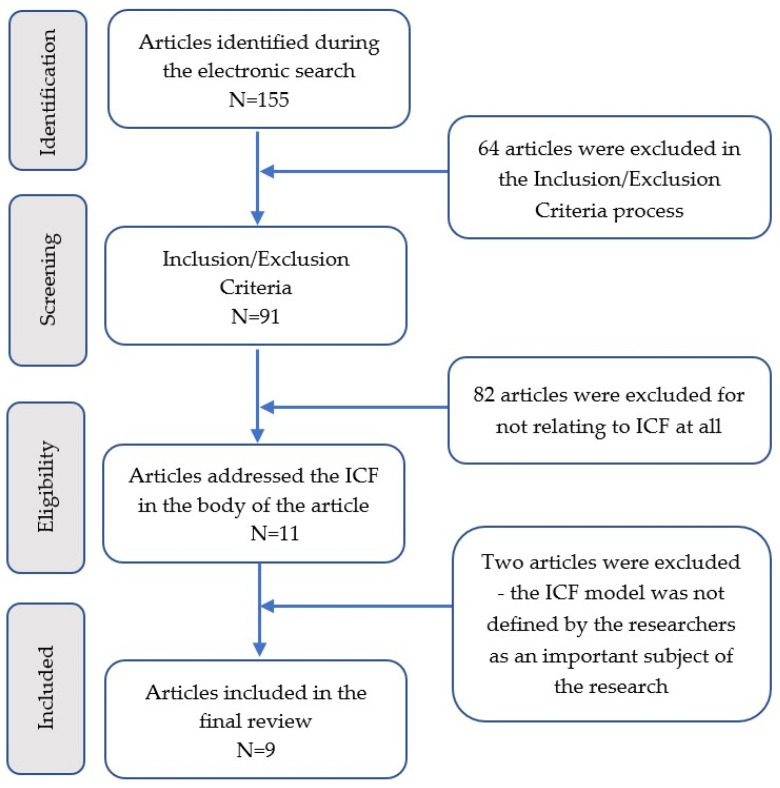
Shows a flowchart of the search and review process (Initial search was completed in January 2020).

**Table 1 children-10-00908-t001:** Initial search criterion keywords.

Hydrotherapy Aquatic Therapy Aquatic Activities Aquatic Exercise Aquatic Exercise Programs Aquatic Fitness Aquatic Group Therapy Aquatic Physical Therapy Aquatic Program Aquatic Sports Aquatic-Based Exercise Program Aquatics Aerobic Aquatic Gymnastics Pool Therapy Pool Therapy Method Swimming Swimming Rehabilitation Swimming Therapy Swimming Training Water Activities Water Based Exercise Water Environment Water Exercise Water Immersion Water Therapy	and	A child/children with developmental delays Aged 6–12	and	ICF

**Table 2 children-10-00908-t002:** Main characteristics of reviewed articles.

First Named Author and Year	Type and Aim	Study Population	Interventions	Measures	Results
Blohm, 2011 [26]	A literature review of 8 articles aimed to review the available evidence regarding the effectiveness of aquatic interventions for children with CP.	113 children or adolescents (ages 3–20) with CP.	Vary from AE sessions 2× weekly to AE combined with swimming sessions.	The ICF-CY framework was used in order to check the outcome measures and results. ICF-CY components examined: BF, A&P.	Ambulatory children and adolescents with CP clearly benefited from aquatic intervention programs in terms of ICF levels—BF, Act. and/or Par. The improvements were sustained several weeks after completion of a program, while others regressed to baseline values. No adverse effects of APT were presented.
Gorter and Currie, 2011 [25]	A literature review of 6 articles aimed to review published literature since 2005, with a focus on AE for children with CP.	45 children and adolescents with spastic CP and other developmental disabilities. Ages 2–21.	Aerobic exercises, strength exercises and other activities that do not fall under any of the above categories.	The ICF-CY framework was used in order to classify the impact of health conditions according to the effect of the ICF-CY components BF, BS, A&P.	Researchers found evidence on effectiveness of AE in children and adolescents with CP are limited. There is a strong potential for aquatic physical activity to benefit children and adolescents with CP.
Cross et al., 2013 [10]	A scoping review of 23 articles aimed to (1) summarize and disseminate the research findings on aquatic interventions for children with disabilities; (2) identify the recurring issues within the pediatric aquatic literature; and (3) investigate the potential utility of ICF as part of the promotion processes of aquatic interventions for children with disabilities.	382 children with disabilities aged 2–12.	Vary from AE, therapy or structured swimming sessions.	The ICF-CY framework was used in order to check the outcome measures and results. ICF components examined: BF, A&P.	The ICF provides a common framework that can enhance communication among aquatic researchers, practitioners, families and policy- and decision-makers, in turn leading to the development of evidence-based aquatic interventions.
Declerck, 2014 [20]	An RCT–cross-over design article that aimed to investigate the effect of swimming on the multiple aspects of functioning at different levels of the ICF framework among ambulant youth with CP.	14 youth with CP, ages 7–17.	A 10-week swimming program in the community. Two sessions per week (30 to 60 min.). All sessions consisted of a 5-to-10 min warm-up, 20 to 40 min of learning new tasks and 5 to 10 min of free play, races and other games.	VAS; Faces Pain Scale—Revised; 10-meter walk test; 1 min fast walk test; PedsQLTM multidimensional fatigue scale; Bruininks–Oseretsky test of motor proficiency; PEDI-NL; Self-Perception Profile for youth with CP; PedsQL™ CP module version 3.0; WOTA 2; 5-point Likert scale; CAPE.	All youth had a high adherence towards the program; they participated in the intervention with high levels of enjoyment, and most youth continued to participate in swimming after completing the program. The intervention had a positive influence on their BF and A&P. One year after the start of the study, they participated in activities of the formal domain and in skill-based activities more with friends and others than with family or alone.
Sutthibuta, 2014 [27]	A systematic review of 3 articles aimed to review the literature according to ICF-CY for clinical applications, further research and practice.	57 children and adolescents with CP aged 6–21.	Different AEs.	Cardiorespiratory endurance; muscle strength; gait analysis; Floor to Stand; PEDI; GMFM; WOTA.	It was shown that the previous studies were not enough to verify the effectiveness of hydrotherapy.
Güeita-Rodríguez et al., 2017 [4]	An experts’ opinion article aimed to identify intervention categories encountered by PTs working in aquatic therapy with disabled children using the ICF-CY.	69 experts (APTs) with experience in AAs for children with disabilities.	The study relied on established linking rules in order to link participants’ responses to the ICF-CY. The ICF-CY language was used to provide a summary of the participants’ answers to questionnaires, and calibration linking was performed by two different health professionals who were trained in ICF-CY linking.	A Delphi consensus process. Response rates were analyzed using descriptive statistics.	A total of 99 ICF-CY categories were identified, which were divided into 4 ICF-CY components as follows: 41 BF, 8 BS, 36 A&P and 14 EF. Regarding the influence of aquatic therapy upon EFs, there was a notable consensus regarding the support, relationships and attitudes of family members.
Khalaji et al., 2017 [28]	An integrative review aimed to review the extant literature in the field of hydrotherapy and its applications for the improvement of ICF in spastic diplegia CP patients.	Children with spastic diplegia CP, aged 4–21.	Different types of hydrotherapy.	The ICF-CY framework was used in order to check the outcome measures and results. ICF components examined: BF, A&P.	Hydrotherapy, when administered with conventional methods for rehabilitation of children and adolescents with spastic diplegia CP, has positive effects on all areas of ICF. The exercises and their duration and intensity should be decided on the basis of the physical and cognitive conditions of the patients.
Güeita-Rodríguez et al., 2018 [29]	An integrative review aimed to explore the experiences regarding aquatic physiotherapy among parents of children with CP and to identify a list of relevant intervention categories for aquatic physiotherapy treatments.	18 parents of children with CP.	Semistructured interviews and focus groups based on the components of the ICF as a frame of reference to explore and code experiences regarding aquatic physiotherapy.	A questionnaire for parents with a topic guide was developed based on the five ICF components. The identified findings of this questionnaire were organized by ICF-CY component and linked to the ICF-CY categories according to established linking rules.	A total of 107 ICF-CY linkages were performed: 42 categories of BF, 12 BS, 42 AP and 11 EF. Parents stressed the importance of AE for their children’s muscle functions and balance as well as for family and social relations and that the current services, systems and health policies represent a barrier for the practice of APT with their children. These results could be included in goal-setting and may enable APTs to develop treatment-based classification systems.
Güeita-Rodríguez et al., 2019 [30]	A consensus process to report on the preliminary APT CS for children and youth with neurological disorders using the ICF-CY version.	15 experts (APTs) who had over 5 years of experience working in water with children and youth with neurological disorders.	A Delphi consensus process that was undertaken in two languages (English and Spanish). The process was completed when a consensus was reached between the experts.	A consensus agreement among the experts was set in a Delphi consensus process.	1. A comprehensive APT CS for children and youth with neurological disorders that included 64 different ICF-CY categories. These 64 categories represent 3.79% of all categories included in the ICF-CY classification; 2. Four brief APT CSs: APT CS aged 9 to 18 years; APT CS for the ages of 0 to less than 6 years; APT CS aged 6 to less than 14 years; APT CS aged 14 to 18 years.

Act: activities component, AEs: aquatic exercises, APT: aquatic physical therapy, APT-CSs: aquatic physical therapy core sets, APTs: aquatic physical therapists, A&P: activities and participation, BF: body function, BS: body structure, CAPE: The Children’s Assessment of Participation and Enjoyment, CP: cerebral palsy, EF: environmental factors, GMFM: gross motor function measure, ICF-CSs: ICF core sets, ICF-CY: International Classification of Functioning, Disability and Health for Children and Youth, Par.: participation component, RCTs: randomized controlled trials, PEDI-NL: Pediatric Evaluation of Disability Inventory—The Netherlands, PedsQL TM: Pediatric Quality of Life Inventory TM, VAS: Visual Analogue Scale, WOTA: Water Orientation Test Alyn.

**Table 3 children-10-00908-t003:** ICF reporting criteria of the nine articles reviewed.

	ICF Reporting Criteria	Blohm [25]	Gorter and Currie [26]	Cross et al. [10]	Declerck [20]	Sutthibuta [27]	Güeita-Rodríguez et al. [4]	Khalaji et al. [28]	Güeita-Rodríguez et al. [29]	Güeita-Rodríguez et al. [30]
1	Are all components of the ICF framework considered?	No	Yes	Partially. All components are mentioned; only 5 of them are referred to.	Yes	Partially. All components are mentioned; only 4 of them are referred to.	No	No	No	No
1.1	Which components are NOT considered?	BS, PF, EF		PF	PF, EF		PF	Par. (may be included in Act.), PF.	PF	PF
1.2	Reasons for excluding components are explained?	No		No		No	PF categories are not classified to date.	No	PF categories are not classified to date.	PF categories are not classified to date.
2	The interactions in ICF are considered? 2.1 ICF interactions are discussed?	No	No	Yes	Yes	Partially. The model was explained, but the focus is only on the social part.	No	No	No	No
3	Demonstrated awareness of the literature predating the study and relevant reference to ICF literature is provided?	Yes	Yes	Yes	Yes	Yes	Yes	Partially. There is no literature reference regarding ICF.	Yes	Yes
4	Explicit references to ICF definitions and categories are included?	No	Yes	Yes	Yes	No	Yes	No	Yes	Yes
5	Consistent use of ICF language is demonstrated?	No. Mixing concepts between ICF components and “Levels”.	No. Mixing concepts between ICF components and “categories”.	Partially. Mixing concepts between ICF domains and “categories”. “Disability” is used incorrectly—”Children with disabilities”.	Partially. Mixing concepts between ICF “domains” and “categories”. “Disability” is used as a word that describes a health status.	Yes	Yes	Partially. Mixing concepts between ICF “domains” and “areas”.	Yes	Yes
6	Where ICF is linked/mapped to another tool, description of the methodology is given?	No	Partially. There are examples of results linked to ICF’s components, but no explanation of the methodology.	Partially. A short description of the matching of outcomes to the ICF components. No explanation of the methodology.	No	No	Yes	Partially. There are examples of results linked to ICF’s components, but no explanation of the methodology.	Yes	Yes
7	If there is linkage between ICF qualifiers and other measures, description of the methodology is given?	No linkage	Partially. Only the term “barriers”. No linkage methodology was given.	No linkage.	No linkage.	No linkage.	Partially. A brief reference on capacity and performance.	No linkage.	Partially. A reference to the term “Barriers”. No linkage methodology was given.	Partially. A brief explanation of the numerical marking method and of “capacity” and “performance”.
8	Description of the use of ICF qualifiers, e.g., five-point scale, three-point scale, present?	No	No	No	No	No	No	No	Partially. Mention the 5-point scale and the term “Barriers” with no description.	Partially. A brief description of the numerical marking method.
8.1	Description of the reason for selection of qualifier use is provided?	No	No	No	No	No	No	No	No	No
9	The person’s perspective is recognized in the reporting?	No	Yes	Not really. Just an incidental mention.	Yes	Not really. Just mention it.	No	No	Yes. With parents’ opinions of their children’s needs.	Maybe. Age-specific ATP-CS groups. For purposes of their use in studies.
10	The relationship between the ICF and an ICF-based instrument is described?	No	No	Yes. The ICF as a tool to classify the various outcome measures.	Yes. The multiple facets of functioning in addition to quality of life.	No	Yes. Using the Delphi consensus process with ICF-CS to identify relevant intervention categories for APT.	No	Yes. The interview guide for parents—abilities of the children’s functioning were linked to the ICF components.	Yes. The APT CSs—a standard of functioning for the use in AT for children and youth.
11	Knowledge translation between different settings is discussed?	No	No	Yes. ICF common language for researchers and interventions.	Yes. ICF linkage to children functioning; ICF as a tool for understanding the relationships between the components found in research.	Yes. The AT activities and QoL is consistent with the ICF framework.	Yes. The Delphi consensus process with standardized WHO methodology for ICF-CSs relevant to APT treatments.	No	Yes. The ICF-CY as a reference for parent interview guide.	Yes. ICF-CY APT- CSs development process that included gathering knowledge from previous studies and an experts’ consensus process based on the Delphi method.

Act: activities component, APT-CSs: aquatic physical therapy core sets, BS: body structure, EFs: environmental factors, ICF: International Classification of Functioning, Disability and Health, ICF-CSs: ICF core sets, ICF-CY: International Classification of Functioning, Disability and Health for Children and Youth, QoL: quality of life, Par.: participation component, PF: personal factor.

## Data Availability

Not applicable.

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
