# Peer review of "The International Classification of Functioning, Disability and Health: Accuracy in Aquatic Activities Reports among Children with Developmental Delay"

_children, 2023, doi:10.3390/children10050908_

Round 1
Reviewer 1 Report
Review Manuscript ID: ijerph-2222510
This reviewer has worked with the ICF and the predecessor ICIDH, giving structure to health care processes. Twenty-two (22) years after publication of the ICF, authors conclude that, despite rising awareness, is still used inaccurately and that a better understanding of ICF is required (see the abstract). The hidden conclusion seems to be that WHO has failed so far to create that understanding in the area of authors of aquatic therapy authors ref aquatic activity in children aged 6-12 with developmental delay.
The authors have created a thorough – and lengthy – manuscript, from which they derive two conclusions: “One is the importance of using the model as a whole, in the appropriate language and without omitting various components which impairs the holism of the framework”, and “The other is researchers' awareness to the impact of the ICF model should play as an evaluation and measurement tool”.
Referring to conclusion one: concrete ideas and plans on how to increase the importance and the knowledge of the model in the aquatic therapy/exercise industry are missing. These would increase the message of the article in increase the justification to be published. A draft of e.g. a reference model, which future authors can support to design their research in aquatic therapy might have been included.
Perhaps conclusion one of the authors is not completely correct. The three articles by Güeita et al were about the design of an aquatic core set, based on ICF terminology; resulting from a PhD thesis, supported by the ICF research branch in Munich (dr Cieza). The thesis explains the model comprehensively, but authors have to adhere to word limits in publications, which certainly is a reason for not being as inclusive as authors of this manuscript would like.
Perhaps publishers should not only ask for article design according to e.g. CONSORT, but also using the ICF framework.
Referring to conclusion two: the word measurement(s) pops up only four time in the text and obviously was of limited importance. Why is it an important conclusion?
The flow diagram shows 82 articles excluded because of not referring to ICF. Eleven (11) were included: authors DID refer to the ICF but did not meet the criteria by Daugaard (2018) completely. Hopefully in future a next article will evaluate reasons why ICF was not used in those 80 excluded articles.
The search in this manuscript needs explanation: Why the comprehensive list of 25 word combinations referring to aquatic activity and only 2 for the population? Why databases as Embase and CINAHL were not included?
Author Response
Dear editors and referees
Thank you for your important remarks.
We made all the changes we could.
Below, 2 tables with your remarks and our answers (red= referee 1; blue= referee 2).
The same colors we used within the revised article.
We hope the changes match what you intended.
Thank you

Reviewer 2 Report
Thank you for the opportunity to read the article “The International Classification of Functioning, Disability, and Health: accuracy in aquatic activities reports among children with developmental delay.”
The topic is very interesting. In addition, the importance of a common language in research is significant. I have a few comments to clarify the article:
· The search range of the articles is between 2010 and 2020. In my opinion, it is worthwhile to expand the search until the end of 2022. It is possible that more relevant articles will be found.
· In some places, there is a reference to children and adolescents, and, in some areas, only to children, but the age is 6 to 12 years without a clear definition of puberty. I suggest using only the term children (without adolescents) throughout the article.
· In the section on inclusion and exclusion criteria, reference the section: Main Population of the studies should be children with developmental delays Aged 6-12. Studies that included different age groups were also included as long as this age group was included in the study. It is not clear enough what is meant by this. Please add an example to clarify.
· Table 3 is not accessible to the reader. It is challenging to understand what it shows or to collect information from it. I suggest thinking of another way to present the information.
· The discussion is very long. I propose shortening it to mainly sections 1,2,4.
Author Response

(The authors gave the same response as above.)

Reviewer 3 Report
Please see the attachment

Round 2
Reviewer 3 Report
Thank you for the careful considerations of comments. The manuscript is more explicit in the methods which is helpful.
The rationale for using the ICF is stronger. Thank you for this.
I suppose what I was asking to be discussed in the introduction is that the ICF model is a model for health professionals (albeit with variable uptake). It a means of creating a common language between health professionals who have their own distinct professional identities and languages. It is not the model of health and wellbeing that most people with lived experience of disability and their families use. In the latter instance, people often use their own cultural models of wellbeing, which very often include spirituality. So, I was hoping you might make this clearer. I was not expecting a comparison of all the various type of health models used in the world.
More clarification of what type of articles included in the review is required. The methods suggest that quasi experimental and various types of reviews are to be included but it looks like a Consensus study was included and a Delphi study (i.e. more of a survey rather than experimental?) was included. Some clearer explanation of this in the methods and then in the results is required.
Most of what was included was reviews. Was this done in preference to including the original studies….? I am wondering if only including reviews would be clearer??? So, a stronger case for how authors of the review link the ICF model across findings from studies might be made?
Or have I misunderstood, and you went back to the original papers included in each of the ‘included’ reviews to make a decision about author mastery?
I suppose I am asking, were you reviewing the mastery of the authors of the 6 reviews included… or are you reviewing the mastery of the original authors.
I am wondering how much overlap there was of the included studies in each of the included reviews? For example, did the same primary study appear in several of the different reviews included in your review?
I again would encourage the authors to review the items in the PRISMA scoping review. These are just concepts to ensure that you have demonstrated strong methodological rigour in your processes. This helps the reader to trust your findings. I am not suggesting you replace the ICF based analysis (as I agree this is appropriate for anything across the studies) but rather that you use the guidelines to ensure you have met the main and applicable reporting categories.
I really like the additions to Table 2. It is now much clearer what data is in your review. thank you.
The discussion is now much stronger. Thank you for considering the main points and then discussing these.
The conclusion is improved but could still be shortened. Ideally no more than 8 sentences should be sufficient to summarize your study.
Author Response
17.5.23
Dear editors and referees
Thank you again for your important remarks.
We made all the changes we could and regarding comments that we did not change - we wrote our reasons in the attached table.
The changes are marked with “Track changes” within the revised article.
We hope the changes match what you intended.
Thank you
|
Remarks & Suggestions |
Our Response |
1. |
Thank you for the careful considerations of comments. The manuscript is more explicit in the methods which is helpful |
Thank you |
2. |
The rationale for using the ICF is stronger. Thank you for this. |
Thank you |
3. |
I suppose what I was asking to be discussed in the introduction is that the ICF model is a model for health professionals (albeit with variable uptake). It a means of creating a common language between health professionals who have their own distinct professional identities and languages. It is not the model of health and wellbeing that most people with lived experience of disability and their families use. In the latter instance, people often use their own cultural models of wellbeing, which very often include spirituality. So, I was hoping you might make this clearer. I was not expecting a comparison of all the various type of health models used in the world. |
We added some explanation in the abstract
|
4. |
More clarification of what type of articles included in the review is required. The methods suggest that quasi experimental and various types of reviews are to be included but it looks like a Consensus study was included and a Delphi study (i.e. more of a survey rather than experimental?) was included. Some clearer explanation of this in the methods and then in the results is required. |
It was not important the kind or design of the study – only whether if it discussed the ICF model and aquatic therapy for children with DD. We added the 2 types of articles we omitted by a mistake. |
5. |
Most of what was included was reviews. Was this done in preference to including the original studies….? I am wondering if only including reviews would be clearer??? So, a stronger case for how authors of the review link the ICF model across findings from studies might be made? Or have I misunderstood, and you went back to the original papers included in each of the ‘included’ reviews to make a decision about author mastery? I suppose I am asking, were you reviewing the mastery of the authors of the 6 reviews included… or are you reviewing the mastery of the original authors. |
We are interested in the content of what has been claimed to be done in the articles regarding the application of ICF in research on AA. |
6. |
I am wondering how much overlap there was of the included studies in each of the included reviews? For example, did the same primary study appear in several of the different reviews included in your review? |
We gave information within the article |
7. |
I again would encourage the authors to review the items in the PRISMA scoping review. These are just concepts to ensure that you have demonstrated strong methodological rigour in your processes. This helps the reader to trust your findings. I am not suggesting you replace the ICF based analysis (as I agree this is appropriate for anything across the studies) but rather that you use the guidelines to ensure you have met the main and applicable reporting categories. |
We appreciate your suggestion, but we are not interested in the PRISMA criteria, which are about the structure of publications and would need an additional review of all the publications. One of the main ideas of this review is to show that the guidelines developed by the developers of the ICF can and should be used when writing and reviewing articles about the ICF framework. |
8. |
I really like the additions to Table 2. It is now much clearer what data is in your review. thank you. |
Thank you |
9. |
The discussion is now much stronger. Thank you for considering the main points and then discussing these. |
Thank you |
10. |
The conclusion is improved but could still be shortened. Ideally no more than 8 sentences should be sufficient to summarize your study. |
We made the changes. |
11. |
English improvement |
we need more days in order to send it – is it OK? |
.
Submission Date
31 January 2023
Date of this review
14 May 2023 21:24:03
